# Hydration Status and Acute Kidney Injury Biomarkers in NCAA Female Soccer Athletes During Preseason Conditioning

**DOI:** 10.3390/nu17132185

**Published:** 2025-06-30

**Authors:** Daniel E. Newmire, Erica M. Filep, Jordan B. Wainwright, Heather E. Webb, Darryn S. Willoughby

**Affiliations:** 1School of Health Promotion and Kinesiology, Texas Woman’s University, Denton, TX 76204, USA; 2Department of Kinesiology, Texas A&M University-Corpus Christi, Corpus Christi, TX 78412, USA; erica.filep@tamucc.edu (E.M.F.); jordan.wainwright5@gmail.com (J.B.W.); 3Department of Exercise Science, Mercer University, Macon, GA 31207, USA; webb_he@mercer.edu; 4Huffington Department of Education, Innovation, and Technology, School of Medicine, Baylor College of Medicine, Temple, TX 76508, USA; darryn.willoughby@bswhealth.org

**Keywords:** acute kidney injury, exercise training, preseason, renal dysfunction

## Abstract

Exercise training in extreme temperatures concurrent with hypohydration status may potentiate the development of acute kidney injury (AKI) in young, healthy persons. **Background/Objectives**: It is unknown how repeated training bouts in ambient higher temperatures and humidity may influence measures of AKI. The purpose of this study was to investigate hydration status and renal biomarkers related to AKI in NCAA Division I female soccer athletes during preseason conditioning. **Methods**: A convenience sample of *n* = 21 athletes were recruited (mean ± SEM; age: 19.3 ± 0.25 y; height: 169.6 ± 1.36 cm; mass: 68.43 ± 2.46 kg; lean body mass: 45.91 ± 1.13 kg; fat mass: 22.51 ± 1.69 kg; body fat %: 32.22 ± 1.32%). The average temperature was 27.43 ± 0.19 °C, and the humidity was 71.69 ± 1.82%. Body composition, anthropometric, workload, and 14 urine samples were collected throughout the preseason training period for urine specific gravity (USG), creatinine (uCr), cystatin C (uCyst-C), and neutrophil gelatinase-associated lipocalin (uNGAL) analyses. **Results**: Our investigation showed that, when compared to baseline (D0), the athletes maintained a USG-average euhydrated status (1.019 ± 0.001) and were euhydrated prior to each exhibition game (D5-Pre: *p* = 0.03; 1.011 ± 0.001; D10-Pre: *p* = 0.0009; 1.009 ± 0.001); uCr was elevated on D8 (*p* = 0.001; 6.29 ± 0.44 mg·dL^−1^·LBM^−1^) and D10-Post (*p* = 0.02; 6.61 ± 0.44 mg·dL^−1^·LBM^−1^); uCyst-C was elevated on D6 through D10 (*p* = 0.001; ~0.42 ± 0.01 mg·dL^−1^); no differences were found in uNGAL concentration. The highest distance (m) displaced was found during exhibition games (D5: *p* = <0.0001; ~8.6 km and D10: *p* = <0.0001; ~9.6 km). During the preseason conditioning, the athletes maintained a euhydrated status (~1.019) via USG, an increase in uCr that averaged within a normal range (208 mg·dL^−1^), and an increase in uCyst-C to near AKI threshold levels (0.42 mg·L^−1^) for several practice sessions, followed by an adaptive decline. No differences were found in uNGAL, which may be explained by athlete variation, chosen time sample collection, and variation in training and hydration status. **Conclusions**: The athletes maintained a euhydrated status, and this may help explain why urinary markers did not change or meet the reference threshold for AKI.

## 1. Introduction

The physiological demands of soccer athletes incorporate both aerobic and anaerobic systems for muscular strength, endurance, power, and agility to optimally perform in 90–120 min matches [1,2]. The National Collegiate Athletic Association (NCAA) only allows 21 practice units (including exhibition games) before the first scheduled date of competition [3]. With a relatively short time frame in the preseason training period, coaches are responsible for preparing student-athletes for the competitive season. During preseason, collegiate soccer athletes complete 2–3 daily sessions, including conditioning, strength, technical, and tactical training, and exhibition games. Geographic location can influence environmental factors that worsen exertional heat illnesses (EHIs), with hot, humid conditions increasing risk [4]. In the southern U.S., NCAA Division I soccer preseason occurs in August, exposing athletes to high temperatures and humidity, raising dehydration risk from increased water loss and insufficient fluid intake [5]. It has been shown that >2% of body mass loss is often associated with an increased risk of EHIs and a decrease in athletic performance [6]. From the archived National Oceanic and Atmospheric Administration (NOAA) data, South Texas experienced an average temperature of 29.78 °C (24.3–35.0 °C) and a relative humidity of 74% during August. These environments can elevate the risk of dehydration, which may lead to severe health complications caused by heat strain from increased core body temperature [5,7]. This, in turn, may further heighten the risk of acute kidney injury [8].

Acute kidney injury (AKI) is defined as a rapid decline in renal function occurring within hours, encompassing both structural injury and functional impairment of the kidneys. AKI is classified into three categories: (a) pre-renal AKI, (b) acute post-renal obstructive nephropathy, and (c) intrinsic AKI. Of these, only intrinsic AKI denotes true parenchymal kidney disease. In contrast, pre-renal and post-renal forms are secondary to extra-renal pathologies that compromise the glomerular filtration rate (GFR). If these conditions persist, they may induce renal parenchymal injury and progress to intrinsic renal pathology [9].

Recent research suggests that acute kidney injury (AKI) can result from strenuous exercise, dehydration, and severe heat illness (e.g., exertional heat stroke) caused by prolonged heat strain and elevated core body temperature [10,11]. Over the past decade, serum, plasma, and urinary biomarkers have been used for early AKI detection [12]. Traditionally, serum creatinine (sCr), a catabolic byproduct of creatine phosphate metabolism in skeletal muscle, has been the primary marker for AKI assessment [13,14]. Approximately 2% of creatine stores are irreversibly converted to creatinine, with levels influenced by lean body mass, age, protein intake, and muscle function. In healthy individuals, creatinine is freely filtered by the kidneys, with minimal extrarenal metabolism. Creatinine is almost entirely excreted in urine, with about 10–30% secreted by the proximal tubules in those with normal basal kidney function [15]. However, exercise can increase urinary creatinine (uCr) excretion without significantly affecting serum levels [13,14,16]. Urinary creatinine (uCr) levels typically range from ~20 to 400 mg·dL^−1^ [17], with higher values indicating potential kidney dysfunction. ICU patients with AKI have reported uCr concentrations of ~133 mg·dL^−1^, compared to ~82 mg·dL^−1^ in non-AKI patients. However, creatinine alone may be an unreliable AKI biomarker, as levels can be affected by diet, malnutrition, exercise-induced muscle damage, and athletes seeking to acquire muscle mass [18]. While creatinine is used to estimate glomerular filtration rate (GFR), it does not specifically indicate a tubular or glomerular injury and may not reliably identify AKI.

The American Society of Nephrology has identified potential biomarkers for AKI detection, including cystatin C (Cyst-C), neutrophil gelatinase-associated lipocalin (NGAL), and creatinine (Cr) [19,20,21]. Cyst-C, a 13 kDa protein, is filtered by the glomerulus, reabsorbed, and metabolized in the renal tubule (Figure 1). Even a slight increase in urinary Cyst-C (uCyst-C) signals proximal tubule injury [12]. As a key extracellular inhibitor of cysteine proteases, Cyst-C is typically reabsorbed in a healthy kidney and absent from urine in significant amounts [22]. It has been reported that the normal reference range for urinary Cyst-C is 0.06–0.16 mg·L^−1^ [23]. However, renal injury reduces reabsorption, leading to an increase in urinary presence. Given that >99% of Cyst-C is filtered and catabolized by the renal tubules, any detectable urinary levels may indicate impairment [19,20]. Previous studies show that in AKI patients, uCyst-C concentrations ≥0.45 mg·L^−1^ suggest tubular dysfunction, while non-AKI levels are ≤0.07 mg·L^−1^. Increased uCyst-C levels appear to be a valid method of detecting tubular dysfunction [24].

Recent studies have explored the effects of exercise on uCyst-C and AKI markers in healthy individuals. Bongers et al. (2018) found uCyst-C levels increased more after prolonged (150 min) than acute (30 min) exercise and rose 1.8-fold after the first day of repeated exercise (0.05–0.09 mg·L^−1^) [25]. However, it was noted that this increase dissipated and returned to near baseline levels after 3 days [26]. Lastly, an increase in uCyst-C was also found in individuals after both 10 and 100 km runs. There was a 2.56-fold increase after 10 km and a 4.96-fold increase after 100 km [27].

In addition to uCyst-C as an AKI measure, neutrophil gelatinase-associated lipocalin (NGAL), also known as human neutrophil lipocalin or lipocalin 2, exists in three molecular forms in blood and urine: a 25 kDa monomer, a 45 kDa disulfide-linked homodimer, and a 135 kDa heterodimer. Synthesized in the bone marrow during myelopoiesis, it is stored in neutrophil granules. NGAL mRNA is also expressed in non-hematopoietic tissues, including the colon, trachea, lung, and kidney epithelium, with synthesis stimulated by Interleukin-1β (IL-1β). In AKI, kidney epithelial cells primarily secrete the 25 kDa monomeric form [28].

Under normal conditions, NGAL is filtered by the glomerulus, reabsorbed in the proximal tubules (Figure 1), and minimally excreted in urine. It has been reported that the normal urinary concentration of NGAL in females is ≤65.0 ng·mL^−1^ [29]. It is produced by activated neutrophils in the proximal tubules. Following ischemic, septic, or toxic kidney injury, NGAL transcription and protein levels rise sharply, increasing plasma and urinary NGAL. Elevated NGAL levels indicate early structural renal tubular damage. It is considered a more specific AKI marker, particularly in healthy individuals, where systemic inflammation and multi-organ damage may otherwise elevate its levels in clinical settings [12,28].

Regarding exercise and AKI, NGAL has been investigated as a potential biomarker. Studies on marathon running found urinary NGAL (uNGAL) concentrations increased from ~8–12 ng·mL^−1^ at baseline to ~33–47 ng·mL^−1^ immediately post-marathon. However, 24 h post-race, one study reported a return to baseline (~10 ng·mL^−1^), while another observed a further rise to ~59 ng·mL^−1^ [30,31]. In shorter duration, higher intensity bouts of exercise, uNGAL concentration changes are variable. Previous studies showed an increase in uNGAL immediately after a high-intensity sprint (800 m) from <10 to ~12 ng·mL^−1^ 25 min post-exercise [32] and after high-intensity interval resistance training (HIIT) and an increase from ~18 to 33 ng·mL^−1^ 2 h post-HIIT [33]. However, similar to some of the observations in the marathon runners, uNGAL concentrations fell back to baseline levels. This may suggest that these AKI markers may be transient, relative to the duration of the exercise bout, and a potential kidney adaptation to exercise stress over time [12].

The purpose of this study was to investigate the roles of athlete workload and hydration status during multiple practice sessions on urinary biomarkers reflective of AKI during preseason training in the summer month of August in NCAA Division I female soccer student-athletes located in South Texas. We hypothesized that the concurrent effect of higher temperatures and humidity, and the assumed preseason training status of the athletes, may negatively affect the markers of hydration and AKI.

## 2. Materials and Methods

### 2.1. Sample Population

This study was approved by the Institutional Review Board (IRB 50–19) and the Institutional Biosafety Committee (IBC) at Texas A&M University-Corpus Christi. Although all convenience samples have less clear generalizability than probability samples, not all convenience samples are the same. Homogeneous convenience samples have clearer generalizability relative to conventional convenience samples with more constrained sociodemographic characteristics [34]. A homogenous convenience sample of 21 female soccer student-athletes competing in NCAA Division I soccer was recruited for participation (age: 19.38 ± 0.25 y; height: 169.6 ± 1.36 cm; mass: 68.43 ± 2.46 kg; lean body mass: 45.91 ± 1.13 kg; fat mass: 22.51 ± 1.69 kg; body fat %: 32.22 ± 1.32%). Athletes were medically cleared to participate by the team physician and athletic training staff to participate in this study. Due to the observational nature of this study, all individuals who were medically cleared to participate in preseason were recruited to participate in the study. Athletes designated for urinary AKI biomarker analysis were selected based on the greatest workload performed over the preseason schedule collected by the Polar Team Pro System.

### 2.2. Anthropometrics and Body Composition

Height and mass were assessed via a portable SECA scale and stadiometer (Seca model 769). Whole body composition, regional body composition, and bone mineral density were assessed using the GE Lunar Dual X-Ray Absorptiometry (iDXA) technology (iDXA, Lunar Prodigy; GE Healthcare, Madison, WI, USA). The iDXA is a relatively quick, safe, and noninvasive method to assess body composition measures [35].

### 2.3. Hydration Status

Hydration status was measured utilizing changes in clothed-body mass pre- and post-practice sessions using the same portable scale and stadiometer (Seca model 769, Seca GmbH & Company, Hamburg, Germany). Urine was collected and analyzed using urine-specific gravity (USG) analysis using a clinical refractometer (Sper Scientific; model 300005; Scottsdale, AZ, USA). The manufacturer reported the accuracy of the clinical refractometer was ±0.002 for USG measurements. Additionally, non-nude body mass was recorded for each participant pre- and post-practice session as a sensitive and simple assessment to determine acute changes in body water for all types of dehydration [36]. Athletes were instructed to wear the same clothing for their pre- and post-practice session body mass measures. Perspiration that may have accumulated in clothing post-practice session was not controlled. For this study, hypohydration was defined as USG >1.020 [37] and >2% of body mass loss [36]. The index for the range of hydration status for USG is listed in Table 1 for reference. Additionally, water ingestion during practice sessions was not controlled or monitored due to the difficulty of observing the daily intake of water in multiple athletes (*n* = 21) over the preseason time period.

### 2.4. Environmental Analysis

All practice and competition events occurred on artificial turf located on the university campus. Since artificial turf can raise surface temperatures by ~10–15 °F, human heat stress can increase, which can adversely affect kidney function [4]. Wet-bulb globe temperature (WBGT) is a measurement of the environment that incorporates radiant heat, humidity, ambient temperature, and wind [39]. As relative humidity rises, the ability to evaporate sweat becomes challenging and can also lead to excess stored metabolic heat from exercise [40]. Utilizing WBGT as an index of human heat strain is the most appropriate environmental measure in this context. Ambient temperature (Td), dry-bulb temperature (Tg), wet-bulb temperature (Tw), and percent relative humidity (%RH) were assessed via both the Kestrel 5000 environmental meter and a Kestrel DROP 2 device (Nielsen-Kellerman Co., Boothwyn, PA, USA) on the playing surface. Dry-bulb temperature (Tg) measures were not enabled on these devices. All dry-bulb data for the training sessions were acquired from the National Oceanic and Atmospheric Administration (NOAA) National Centers for Environmental Information (NCEI) [41]. The Corpus Christi Naval Air Station (NAS) served as the closest weather station (4.1 miles) to the practice and competition field. Due to this limitation, WBGT calculations are approximations. After the average height of the athlete population was determined, environmental measures were collected at ~15 cm above the turf with the Kestrel DROP 2 device and at the average one-half height among athletes’ Anterior Superior Iliac Spine (ASIS). Environmental data were continually collected using the Kestrel 5000 unit starting 30 min prior to each outdoor practice and exhibition match. Environmental sampling and storage of data automatically occurred every 15 s. Each practice session was approximately 97 min in duration. Wet-bulb globe temperatures (WBGTs) were calculated using the following equation: 0.7 Tw + 0.2 Tg + 0.1 Td, where Tw = wet-bulb temperature, Tg = globe (dry bulb) temperature, and Td = ambient temperature [42]. Since dry-bulb measurements were not assessed on-site with the environmental data loggers, the dry-bulb data were pooled from the NOAA weather station at Corpus Christi NAS for the preseason period at the times when practices and competitions occurred. All other on-site measures were utilized in the WBGT equation. Due to this limitation, measures are approximations since dry-bulb measures were not enabled on the devices. WBGT approximations were 27.43 ± 0.19 °C and 71.69 ± 1.82 %RH during the intervention.

### 2.5. Athlete Workload Data Collection

The athletes were provided with and fitted for a Team Polar Pro monitor, which utilized a 10 Hz global navigation satellite system (GNSS) device tracked using the Polar Team Pro System (Polar Electro Co., Kempele, Finland). This 10-Hz system has been shown to be an accurate and reliable device for tracking team sports work variables [43]. However, it should be highlighted that this system inherently uses the commonly used “HR_Max_ = 220 − Age”, which has been suggested to be inaccurate with a standard deviation of 10–12 bpm and has been shown to underestimate HR_Max_ in younger and older adults [44]. Workload metrics to highlight the subject workload per session were total distance (m), average velocity (m·min^−1^), age-predicted heart rate average (APHR_Avg_ %), and max (APHR_Max_ %). The Polar Team Pro System was owned and operated by the Athletics Department.

### 2.6. Data Collection Procedures

Athletes were asked to provide 14 urine samples during preseason training, which included two exhibition games. Prior to arriving, athletes were informed of training protocols and were provided consent forms via email. After consenting to participation, athletes were then scheduled between 05:00 and 07:00 on D0 (Table 2). Athletes were then asked to report to the laboratory for baseline anthropometrics and urine collection assessments.

On the day of baseline measures (D0), height and mass were measured using a portable SECA scale and stadiometer (Seca model 769), and then body composition analysis was conducted utilizing dual X-ray absorptiometry (DXA) technology (iDXA, Lunar Prodigy; GE Healthcare, Madison, WI, USA). Athletes were provided with a urine specimen cup and directed to the nearest restroom to the laboratory. After baseline collections on Day 0 (D0), preseason training began the following morning around 07:00. This included the start of fitness testing assessments. Fitness testing assessments were conducted over the first two days of preseason (Days 1 and 2; D1 and D2). Urine samples were collected each morning between 05:00 and 06:30 throughout the preseason training period. Due to the physiological demands and potential lack of heat acclimation upon the start of the preseason, urine collection occurred every morning during the first week of the preseason to closely monitor hydration status and renal function (following D1 and D2). To assess the impact of exhibition games (D5 and D10) on renal function, urine samples were collected prior to the initiation of game warm-ups (Pre-D5 and D10), post-game (Post-D5 and D10), 12 h (D5-12 h and D10-12 h), and 24 h (D5-24 h and D10-24 h) post-exhibition games. Morning urine collection continued at 05:00–06:30 prior to regular preseason training (D8 and D12). Ending measurements were collected at 05:00–06:30 prior to the athletes traveling out of the area (D14) (see Table 1). Pre–post-practice session mass was collected to assess changes in hydration status following exercise in the heat. Due to the limited time availability to collect mass and urine samples during the practice sessions, it was not feasible to collect body mass measures. It has been suggested that 3 serial mass measures should be collected and averaged [45] for greater accuracy. At the end of the preseason (D14), the athletes were asked to report to the laboratory for ending body composition measures and urine collection.

### 2.7. Urinary Collection and Analysis

Urine collection primarily took place in the morning during designated practice and peri-exhibition games (see Table 2 for specific times). The athletes were asked to provide a mid-stream sample after using a cleansing towelette. Specimen cups were placed in the corresponding athlete locker each morning of designated urine collection (i.e., regular practice day) and handed to the athletes the night prior if they could not wait until arrival in the locker room to provide their sample. Athletes arrived at the locker room and provided a morning urine sample. Following the exhibition games (D5 and D10), athletes were granted rest days (D6 and D11) by their coaching staff and were given labeled urinary specimen cups for 12 h post-exhibition games. The athletes were instructed to bring these morning urine samples with them as they returned to practice, and they were placed inside a designated cryo-cooler for transportation to the laboratory for cryostorage (−80 °C) and later analysis.

Urine collection during the exhibition games took place immediately prior to the warm-up and within 30 min following the completion of the game. Specifically, D5 pre-game urine was collected at 15:00 h (D5-Pre) and 19:00 following the completion of the game (D5-Post). Similarly, on the 2nd exhibition game day, D10-Pre urine was collected at 17:00 h and 21:00 h D10-Post. Upon entering the locker rooms, the athletes were given a labeled urine specimen cup and asked to place their samples in a designated cooler. Following the completion of the game, the athletes were provided another labeled urine specimen upon re-entering the locker room. The athletes were asked again to place their samples in a designated cooler for transportation back to the laboratory for cryostorage (−80 °C) and later analysis.

### 2.8. Urine Specific Gravity (USG)

Assessment of USG occurred throughout preseason training with a digital clinical refractometer (Sper Scientific; Model 300036; Scottsdale, AZ, USA). Following a similar handheld digital refractometer protocol [46], urine samples designated for USG were stored at −20 °C throughout the preseason data collection and were analyzed upon completion of data collection. Samples were removed from −20 °C and brought to room temperature (20–25 °C) prior to analysis of USG. Urine samples were measured in duplicate and compared to deionized water (ddH_2_O) and a known control of NaCl (1.020). On baseline (D0) and ending day (D14), samples were collected in the nearest restroom by the laboratory, and USG was assessed following the completion of data collection. On practice days (D1, D2, D8, and D12), urine was collected in the locker room near the practice field stadium. Following collection, urine samples were transported back to the laboratory in a cryo-cooler and stored at −80 °C for future analysis.

### 2.9. Urinary Markers of AKI: Cystatin C, NGAL, and Creatinine

The athletes with the highest workload ratio (m·min^−1^·HR_Avg_^−1^) collected from the Polar Monitor Pro System were selected for urinary analysis for AKI markers. It was assumed that these athletes would have had the highest stress placed on their inherent renal system due to higher average workloads during the preseason. Prior to analysis, samples were thawed to room temperature (~25 °C) and centrifuged for 10 min at 12,000× *g*, and the supernatant was extracted for analysis. Urinary Cyst-C was analyzed utilizing the Human Cystatin C Platinum ELISA Kit (Invitrogen, Carlsbad, CA, USA: BMS2279) where Cyst-C binds to HRP-conjugated anti-human Cyst-C antibody, allowing quantification of Cyst-C following a similar protocol that has been previously published [47]. Urinary NGAL was assessed with an NGAL (KIT 036, BioPorto Diagnostics, Hellerup, Denmark) sandwich ELISA performed in microwells coated with a monoclonal antibody to human NGAL following a similar protocol that has been previously published [48]. Urinary creatinine was evaluated via a colorimetric Urinary Creatinine Assay Kit (Cell BioLabs, Inc., San Diego, CA, USA: STA-378). All ELISA kits followed manufacturer recommendations and were read using a Bio-Rad iMark microplate absorbance reader (Bio-Rad Laboratories Inc., Hercules, CA, USA). Intra-assay CV% was found to be 9.81%.

### 2.10. Statistical Analysis

Data are expressed as mean ± SEM and 95% confidence intervals (95% CIs) where appropriate. However, descriptive data are expressed in mean ± SD and 95% confidence intervals (95% CIs) or in min to max to better express the homogeneity of the sample population. While there is no consensus, the normalization of urinary biomarkers with creatinine in clinical settings is a standard practice to control for variations in urine concentration and is recommended for spot urine sample collection [49]. Therefore, urinary markers of AKI were expressed in both absolute and normalized values to compare concentration to standardized reference values and control for potential urine concentration variation. Additionally, creatinine was expressed in both absolute and normalized concentrations to control for any potential variation in lean body mass (LBM) [49]. All practice session workload and urinary analysis (uCyst-C, uNGAL, and uCr) were analyzed using a one-way repeated-measures (RM)ANOVA or a mixed-effect model when appropriate to observe differences during the preseason. The mixed-effect model uses a compound symmetry covariance matrix and is fit using restricted maximum likelihood (REML). In the absence of missing values, this method gives the same *p*-values and multiple comparison tests as RMANOVA. In the presence of missing values (missing completely at random), the results can be interpreted similarly to RMANOVA. Sphericity was not assumed; therefore, Geisser–Greenhouse correction was used. A post hoc analysis was performed with either Tukey for multiple comparison tests or Dunnett for comparing baseline (D0) to other practice session days when appropriate. Unpaired Student’s *t*-tests were used when appropriate to compare and highlight average differences found in the corresponding post hoc analysis. Where appropriate, correlational analyses were conducted to examine the relationships between workload markers (e.g., distance) and physiological indicators of hydration status (e.g., USG) and AKI biomarkers (e.g., uCyst-C, NGAL). Pearson or Spearman correlation coefficients were calculated based on data normality to assess the strength and direction of these associations. This analysis aimed to determine if training load was associated with changes in hydration or renal stress markers. Equal variance was not assumed, and therefore, Welch’s correction was used. All statistical analysis and figure fabrication were completed with GraphPad Prism version 10.2.3 (San Diego, CA, USA).

## 3. Results

### 3.1. Athlete Workload

The average workload for each variable acquired for each subject can be found for each practice session in Figure 2. Main effect differences in workload were found between practice sessions in APHR_Avg_% (*p* = <0.0001 ****), APHR_Max_% (*p* = <0.0001 ****), distance (*p* = <0.0001 ****), and velocity (*p* = <0.0001 ****). The highest APHR_Avg_% was found at the beginning of the season, while some of the lower values were found during AM practice sessions and near the end of the season. The highest HR_Max_% found was on D1 and D7 (>100% of APHR_Max_, respectively). The greatest distance displaced during the preseason was found during exhibition game 1 (D5PM2; 8680 ± 710.8 m; 95% CI: 7193–10,168 and exhibition game 2 (D10PM; 9670 ± 553.2 m; 95% CI: 8512–10,828 m). Comparing the average total distance during practice sessions and the exhibition games, there was a higher average total distance displaced during the exhibition games (D5PM2 and D10PM; *p* = 0.007 **): practice sessions: *n* = 16; 3164 ± 345.7 m; 95% CI: 2427–3901; exhibition games: *n* = 2; 9175 ± 495 m; 95% CI: 2885–15,465 m (Figure 2). No distance difference was found between the exhibition games. The greatest velocity measures (m·min^−1^) were found on D1 (~69 m·min^−1^) and D7 (~107 m·min^−1^). Figure 3 shows that the time spent in the categorical HR range of 50–59% (*p* = 0.002 **; 18.58 ± 1.3%; 95% CI: 15.83–21.34%), 60–69% (*p* = <0.0001 ****; 19.76 ± 1.03 %; 95% CI: 17.56–20.02%), 70–79% (*p* = 0.0004 ***; 17.41 ± 1.23%; 95% CI: 14.79–20.02%), and 80–89% (*p* = 0.0001 ***; 16.53 ± 1.93%; 95% CI: 12.43–20.62%) was greater than 90–100% (6.86 ± 1.40%; 95% CI: 3.89–9.84%). This indicates that during each practice session, the athletes spent the least amount of time in the 90–100% APHR_Max_ range (*p* = <0.0001). No simple time effect differences were found between other APHR_Max_ ranges (Table 3).

### 3.2. Hydration Status

A main effect difference was found in body mass change during each practice session (*p* = <0.0001). The greatest difference in pre- and post-body mass measures was found during the exhibition games D5PM (−1.16 ± 0.12 kg; 95% CI: −1.41–0.90 kg) and D10PM (−1.40 ± 0.14 kg; 95% CI: −1.71–1.01 kg) where the athletes lost −1.1 kg (Figure 4). The mean average mass loss over the preseason was −0.67 ± 0.06 kg; 95% CI: −0.81–0.53 kg. No differences in body mass loss were found between practice sessions and exhibition games (*p* = 0.07). The USG analysis showed a main effect of time on USG (*p* = <0.0001 τ). Post hoc analysis showed that D5-Pre (1.012 ± 0.001 g·mL^−1^; 95% CI: 1.008–1.016 g·mL^−1^; *p* = 0.04 *) and D10-Pre (1.009 ± 0.001 g·mL^−1^; 95% CI: 1.006–1.012 g·mL^−1^; *p* = 0.0003 ***) had lower USG values than baseline (D0). Comparing the average USG measures between the practice session and the exhibition games showed a difference (*p* = 0.04) and lower average USG value for pre-exhibition game days (D5-Pre; D10-Pre) (*n* = 2; 1.011 ± 0.001 g·mL^−1^; 95% CI: 0.991–1.03 g·mL^−1^) compared to average practice sessions (*n* = 12; 1.020 ± 0.000 g·mL^−1^; 95% CI: 1.018–1.022 g·mL^−1^) (Figure 5).

### 3.3. Urinary Markers of Acute Kidney Injury

The analysis of absolute uCr concentration (mg·dL^−1^) showed a main effect of time on uCr (*p* = <0.0001) where uCr concentration was elevated compared to baseline (D0) on D8 (274.8 ± 17.23 mg·dL^−1^; 95% CI: 236.9–312.7 mg·dL^−1^; *p* = 0.002 **) and D10-Post (293.9 ± 23.27 mg·dL^−1^; 95% CI: 242.7–345.1 mg·dL^−1^; *p* = 0.02 *). The average absolute uCr concentration (mg·dL^−1^) was found higher (*p* = 0.001 **) comparing the peak uCr in practice sessions D8 and D10-Post (*n* = 2; 284.4 ± 9.55 mg·dL^−1^; 95% CI: 163–405.7 mg·dL^−1^) to the other practice sessions (*n* = 12; 195.2 ± 15.23 mg·dL^−1^; 95% CI: 161.7–228.8 mg·dL^−1^). Similar to the absolute uCr analysis, differences were found at D8 and D10-Post in the normalized (mg·dL^−1^·LBM^−1^) analysis; a main effect of time difference was found uCr (*p* = <0.0001) with elevated uCr at D8 (6.29 ± 0.56 mg·dL^−1^·LBM^−1^; 95% CI: 5.31–7.27 mg·dL^−1^·LBM^−1^; *p* = 0.001 **) and D10-Post (6.61 ± 0.38 mg·dL^−1^·LBM^−1^; 95% CI: 5.76–7.46 mg·dL^−1^·LBM^−1^; *p* = 0.02 *) compared to baseline (D0). The average normalized uCr concentration (mg·dL^−1^·LBM^−1^) was found higher (*p* = 0.0002 ***) in the practice sessions D8 and D10-Post (*n* = 2; 6.45 ± 0.16 mg·dL^−1^·LBM^−1^; 95% CI: 4.42–8.48 mg·dL^−1^·LBM^−1^) compared to the other practice sessions (*n* = 12; 4.38 ± 0.34 mg·dL^−1^·LBM^−1^; 95% CI: 3.62–5.13 mg·dL^−1^·LBM^−1^) (Figure 6).

The analysis of absolute uCyst-C concentration (mg·L^−1^) showed a main effect of time on uCyst-C (*p* = 0.001) with elevations in uCyst-C at D6–12 h (0.41 ± 0.02 mg·L^−1^; 95% CI: 0.35–0.48 mg·L^−1^; *p* = 0.0001 ***), D6–24 h (0.40 ± 0.02; 95% CI: 0.34–0.46 mg·L^−1^; *p* = <0.0001 ****), D8 (0.41 ± 0.04 mg·L^−1^; 95% CI: 0.30–0.52 mg·L^−1^; *p* = 0.002 **), D10-Pre (0.40 ± 0.02 mg·L^−1^; 95% CI: 0.34–0.46 mg·L^−1^; *p* = <0.0001 ****), and D10-Post (0.48 ± 0.08 mg·L^−1^; 95% CI: 0.27–0.69 mg·L^−1^; *p* = 0.008 **) compared to baseline (D0). The average absolute uCyst-C concentration (mg·L^−1^) was found to be higher (*p* = <0.0001 ****) in the practice sessions D5-Post through D10-Post (*n* = 6; 0.42 ± 0.012 mg·L^−1^; 95% CI: 0.39–0.46 mg·L^−1^) compared to the other practice sessions (*n* = 8; 0.12 ± 0.03 mg·L^−1^; 95% CI: 161.7–228.8 mg·L^−1^). Similarly, the analysis of normalized uCyst-C (ng·mgCr^−1^) showed a main effect of time (*p* = <0.0001) with an elevation in uCr concentration at Day 5-Post (1460 ± 532.9 ng·mgCr^−1^; 95% CI: 155.8–2764 ng·mgCr^−1^; *p* = 0.04 *), D6–12 h (1989 ± 221.5; 95% CI: 1478–2500 ng·mgCr^−1^; *p* = 0.002 **), D6–24 h (3143 ± 485.5 ng·mgCr^−1^; 95% CI: 1995–4291 ng·mgCr^−1^; *p* = 0.008 **), D8 (1657 ± 213.2 ng·mgCr^−1^; 95% CI: 1165–2148 ng·mgCr^−1^; *p* = 0.003 **), D10-Pre (4413 ± 583.9 ng·mgCr^−1^; 95% CI: 3033–5794 ng·mgCr^−1^; *p* = 0.002 **), and D10-Post (1650 ± 248.4 ng·mgCr^−1^; 95% CI: 1062–2237 ng·mgCr^−1^; *p* = 0.004 **) compared to baseline (D0). The average normalized uCyst-C concentration (ng·mgCr^−1^) was found higher (*p* = 0.01 *) in the practice sessions D5-Post through D10-Post (*n* = 6; 2385 ± 475 ng·mgCr^−1^; 95% CI: 1164–3606 ng·mgCr^−1^) compared to the other practice sessions (*n* = 8; 565.1 ± 117.1 ng·mgCr^−1^; 95% CI: 288.2–842 ng·mgCr^−1^) (Figure 7). No differences (*p* > 0.05) were found comparing any practice day to baseline (D0) in either absolute (mg·mL^−1^) or normalized (ng·mgCr^−1^) uNGAL concentration during the preseason (Figure 8).

## 4. Discussion

In this study, we observed the impact of preseason training on hydration status and renal stress biomarkers that indicate AKI in NCAA D1 female soccer players exposed to elevated temperatures and humidity normally found in South Texas in August. To our knowledge, only one previous study has assessed renal function in soccer players post-match play [50]. This is the first study to follow and examine multiple practice sessions and the hydration status of female soccer athletes and how this may impact urinary-derived biomarkers associated with AKI. In our observations, we found that these athletes maintained a euhydration status over the 2 weeks, yet they were in a greater hydration status prior to their exhibition games. Additionally, we observed an increase in the urinary AKI marker uCyst-C following the first exhibition game near the reference threshold for AKI.

### 4.1. Athlete Workload

The workload markers were used as a proxy to isolate athletes who may have the highest potential risk of acquiring an AKI and to determine if any marker may be interrelated to AKI incidents. However, we did not find an association between distance workload and outcome measures of AKI and USG. Additionally, due to the inherent use of APHR_MAX_ of 220-age by Polar Team Pro that was used to predict HR_Max_ for the athletes and its known inaccuracies [44], great caution should be taken when interpreting and generalizing the HR results of this study. Concurrent with the limitations of the inherent HR_MAX_ prediction equation, variability can be found in heart rate in athletes [51], cardiovascular adaptations may have taken place during the short preseason, and the player’s position, which may limit workload and increase variability (e.g., goalie). During the majority of the preseason, the athletes maintained within an APHR_Max_ range of 50–89% intensity. A limited amount of time was spent in 90–100% of the preseason training period. Notably, a dramatic increase in distance was displaced during the two exhibition games (D5 ~8.6 km and D10: 9.6 km). Previous reports observing soccer athlete displacement during women’s soccer game matches were ~5.4 [52], 5.6 km [53], and 8.9–9.1 km [54], which was similar to our findings. While the increased distance displaced during the exhibition game play was to be expected, what is interesting is that the increase in recorded distance displaced on D5 was followed by the increase in the AKI marker uCyst-C by ~81% (~0.04 to ~0.4 mg·L^−1^) and maintained this increase over several following days (D6–D10). However, this increase in uCyst-C was followed by a sharp decrease in uCyst-C following the second exhibition game (D10), which may suggest some renal adaptation to the stress of exercise training. A previous study that examined the effect of distance in 10 and 100 km marathon racing on uCyst-C concentration values found an increase of ~49% and ~66% from pre- to post-marathon in each distance, respectively [27].

### 4.2. Hydration Status

Hypohydration is defined as a body water deficit that is greater than normal fluid fluctuations. Body water deficits >2% of body mass relative to plasma volume reductions and an increase in plasma osmolarity are proxy measures of hydration status. Both body mass change and urine specific gravity (USG) were used to assess body water changes over the preseason time. Roughly, a >2% of body mass loss is related to a ~3% loss in body water and a deficit [36]. Our analysis showed that the athlete’s mean average pre- and post-practice session body mass loss was <1 kg (−0.67 ± 0.06 kg), which equated to <1% of body mass loss. This outcome parallels the mass loss reported in similar-aged female soccer players <1 kg (−0.40 ± 0.52 kg) that were observed in hotter environments in August. However, it should be highlighted that previous work controlled for fluid intake, urine output, and estimated sweat rate [55]. Additionally, due to the inherent ‘field-based’ study design, we opted to collect non-nude body mass, and the athletes were not instructed to change into dry clothes or towel off excess perspiration post-practice sessions, which has been a recommended protocol to more accurately measure body mass changes [56]. Because we did not control for these factors, we are unable to definitively suggest that these athletes were within a normal or abnormal fluctuation of body water. The use of USG has been a well-established and widely accepted method for evaluating hydration status. It offers a noninvasive and easily accessible measure of urine concentration, with higher values indicating dehydration and lower values reflecting adequate hydration. However, while USG is generally reliable, it has certain limitations influenced by individual factors and should be interpreted alongside other clinical assessments for a more comprehensive evaluation [57]. Our USG data indicate that during the preseason overall, at the time points samples were collected, the athletes maintained an average hydration state of 1.019 ± 0.001 g·mL^−1^, which indicates that, overall, they maintained euhydration. The USG values indicate that athletes arrived at exhibition games (D5-Pre and D10-Pre) in a euhydrated state (1.01 ± 0.001 g·mL^−1^; 95% CI: 1.007–1.015 g·mL^−1^ and 1.009 ± 0.001 g·mL^−1^ 95% CI: 1.006–1.012 g·mL^−1^, respectively) and maintained a normal euhydrated state during the other practice sessions. It should be noted that the athlete exhibition games were in the evening, which may have allowed more time to consume fluid compared to morning practice sessions. The USG-assessed outcome observed in our study is similar to a previous study examining soccer training in the hotter month of August in similar-aged females, where they reported a normal USG-derived hydration value of 1.20 ± 0.01 g·mL^−1^ [55]. In contrast, other studies have highlighted that hypohydration is a common outcome in the assessment of athletes. The increase in USG values following practices is a common occurrence amongst athletes. It was highlighted that 89.8% (53 out of 59) of athletes were categorized as hypohydrated following their respective training with USG measures. It was also noted that the athletes maintained this hypohydrated state despite fluid availability in training facilities during practices [58]. Another study assessed the difference between indoor and outdoor training on hydration status. The authors found that when exercising indoors, 49% of athletes were hypohydrated, increasing to 58% when the same group exercised outdoors. However, temperatures indoors (19–25 °C) were higher than those recorded outdoors (17–23 °C). The increase in the prevalence of hypohydration was related to reduced fluid intake when training outdoors in comparison to indoors. The authors attribute the higher fluid intake to the elevated indoor temperatures and the potential increase in circulating vasopressin, although not directly assessed [59]. It is known that potential increases in serum osmolality and sodium (Na^2+^) concentrations stimulate the secretion of the antidiuretic hormone arginine vasopressin (AVP) in an attempt to increase water reabsorption and maintain homeostasis through the sensation of thirst [60]. However, our study did not assess circulating vasopressin, and its role is solely speculative. It has been documented that first-morning urine samples may be more susceptible to heightened USG scores due to increased urinary concentrations overnight [58]. Lastly, it should be noted that daily dietary and water intake were not recorded for this ‘field-based’ study. Due to this limitation, we are unable to isolate and explain how that may influence hydration status in these athletes.

### 4.3. Urinary Markers of AKI

Creatinine concentrations are largely influenced by skeletal muscle, as creatinine concentration may be relative to the usage and storage of creatine phosphagen in muscle [61,62,63]. However, previous research has revealed that uCr may be expressed in concentration without the influence of lean body mass (LBM) on measures. For ease of comparison and to control for individual skeletal muscle mass, the results of uCr analysis were expressed in concentration (mg·dL^−1^) and normalized by the athlete’s LBM (mg·dL^−1^·LBM^−1^). Increases in uCr have also been associated with an increased glomerular filtration permeability and saturation of filtered proteins via proximal tubular reabsorption [63]. It has been suggested that strenuous physical activity in a hot environment has been associated with acute renal impairment due to reductions in renal blood flow and increased protein excretion due to muscular damage (i.e., nucleotide release). Although exercise-induced damage was not directly evaluated in this study, several authors have suggested uCr as a potential indicator of muscle damage [10,32]. Notably, a rather constant rate of creatinine is excreted in urine, as 15–20% of uCr is actively secreted from the blood through renal tubules in an attempt to maintain normal serum creatinine levels [14]. Thus, abnormal increases in uCr may indicate an increased utilization of creatine phosphate and/or glomerular permeability if renal injury is present. When utilizing non-normalized uCr concentration for comparison, our study indicated a higher concentration of uCr on D8 (*p* = 0.002; 274.8 ± 17.23 mg·dL^−1^; 95% CI: 236.9–312.7 mg·dL^−1^) and D10 post-exhibition game 2 (*p* = 0.02; 293.9 ± 23.27 mg·dL^−1^; 95% CI: 242.7–345.1 mg·dL^−1^), which are values similar to individuals assessed with renal dysfunction [14]. However, it is more likely that the increase in uCr concentration found during the mid-portion of the preseason is in response to an increase in creatine phosphagen turnover and metabolism from high-intensity practice sessions, muscle-induced injury, and glomerular filtration permeability [10,61,63].

In the analysis of the normalized uCr concentration during the preseason practice sessions, it was found that the practice sessions D8 and D10-Post had the highest uCr values (*p* = 0.001; 6.29 ± 0.44 and *p* = 0.02; 6.61 ± 0.38 mg·dL^−1^·LBM^−1^) when compared to baseline. No differences were found between baseline and pre-exhibition game uCr values. It is unknown what the athlete’s training status and activity level were prior to the beginning of preseason training, which could potentially perturb values that were considered normal comparative values. Additionally, dietary and supplementary intake were not recorded or controlled for this study. It has been shown that creatinine clearance increases the more that the daily protein intake increases [64].

Variable responses to exercise in uCr are commonly reported in the literature. Previous investigations assessed uCr following an acute and prolonged bout of cycling and found that in an acute bout of exercise, uCr increased to 104.07 mg·dL^−1^, while a prolonged bout increased uCr to 297.5 mg·dL^−1^ [25]. Similarly, Turgut et al. (2003) assessed uCr concentrations following 2 h of exercise [63]. In female athletes, uCr increased from 98.74 to 201.55 mg·dL^−1^ following exercise. Notably, male athletes had significantly higher uCr levels than female athletes. The differences can be attributed to males having greater muscle mass than their counterparts [63]. Thus, it is evident that skeletal muscle mass influences uCr concentration measures. It may be assumed that the increase in uCr may be attributed to the amount of skeletal muscle mass and the metabolism of creatine phosphate stores in response to strenuous physical activity rather than potential renal dysfunction [61,62].

Cystatin C is normally filtered by the glomerulus, reabsorbed, and metabolized in the renal tubule. A small elevation of urinary Cyst-C (uCyst-C) may reflect proximal tubule injury [12,65]. Nejat et al. (2010) defined AKI and tubular dysfunction as a uCyst-C concentration of ≥0.45 mg·L^−1^ [66]. Our data indicated that the athletes investigated (*n* = 9) remained within normal ranges (0.03–0.3 mg·L^−1^) [24,67] at baseline through D5 (Exhibition game 1). However, following D5, uCyst-C values rose to near the AKI-defined threshold (~0.42 mg·L^−1^) for several practice session days (D5–D10) until concentration values began to decline. This outcome suggests that there was a potentially exercise-induced proximal tubular stress following the first exhibition game that lasted for several practice sessions. This was followed by a reduction in uCyst-C, which may suggest a renal adaptation to this stress. This “renal adaptation” was found to be similar to what was observed and reported previously [26].

To our knowledge, this study is one of the first to observe uCyst-C concentration over several days. Our data reveal that uCyst-C concentrations did not significantly increase until after the first exhibition match (D5) and remained elevated beyond normative values until following the second exhibition match (D10). It was assumed that the athletes were in a “less trained status” before the beginning of the preseason. However, because the training status of the athletes was unknown, this is merely speculation. It is unclear why there was no meaningful change in uCyst-C during the first few practice sessions, and it began to elevate following the first exhibition game (D5).

There are limited investigations that have observed the impact of exercise training on uCyst-C concentration in athletic populations. Previous work assessed the impact of acute versus repetitive moderate-intensity exercise on renal injury. Baseline values of uCyst-C (0.05 mg·L^−1^) increased following an acute bout of walking (0.09 mg·L^−1^). However, it declined following the third consecutive day of walking (0.06 mg·L^−1^) [26]. However, uCyst-C concentrations were within normal ranges, and it was concluded that the exercise intensity did not influence tubular dysfunction. Another study assessed uCyst-C following an acute (30 min) and prolonged (150 min) bout of exercise concurrent with a ~3% dehydrated status. The authors found an increase in uCyst-C in both conditions (acute: 0.03 mg·L^−1^; prolonged: 0.15 mg·L^−1^); however, a higher uCyst-C was observed following prolonged exercise [25]. This suggests that uCyst-C may be sensitive to both the duration and intensity of exercise [26]. Lastly, these same authors found an increase in uCyst-C following a 10 and 100 km run. They observed an increase from 0.04 to 0.11 mg·L^−1^ following the 10 km run and an increase from 0.02 to 0.13 mg·L^−1^ following the 100 km run. Similar to the studies aforementioned, these values fall within a normal concentration range of uCyst-C [27]. However, the authors suggested that uCyst-C may be a very sensitive marker of proximal tubule dysfunction after exercise [12].

These studies examined the impact of exercise duration, intensity, and hydration status on uCyst-C concentration. However, the impact of repeated daily bouts of strenuous exercise may further impact uCyst-C concentrations and further kidney injury. Our findings indicate that uCyst-C may remain elevated due to repeated bouts of strenuous physical. It is unknown if the temperature of the athletes exposed exacerbated this response. While it is interesting that uCyst-C concentration values increased after D5, which was recorded as the highest ambient temperature during the preseason (36.6 °C) and one of the highest distances displaced during the preseason (*p* = <0.05; 8680 ± 710.8 m; 95% CI: 7193–10,169 m), it is unknown if the rise in uCyst-C is directly related or interrelated to the response to training status, exercise intensity, duration, repetitive practice sessions, hydration status, and exposure to extreme temperatures. Interestingly, a small sample of the total athletes observed (*n* = 9) at differing time points between practice sessions D5 and D10 showed uCyst-C values higher than what is considered the threshold for AKI (≥0.45 mg·L^−1^). This may suggest there may be inter-variability in the renal response to exercise-induced stress in that some may be more prone to a potential AKI than others. However, extreme caution should be exercised when suggesting that uCyst-C values near the threshold are indicative of a significant AKI due to the subsequent reduction after D10. Previous authors have suggested normalizing uCyst-C measures with other metabolites, such as creatinine, may be more relevant to an accurate uCyst-C concentration [12]. The normalization of AKI biomarkers to uCr may account for any uncontrolled dilution-related variations (e.g., dietary intake, physical activity levels, kidney function, and medication) and compare biomarker levels more accurately across different individuals or time points [49]. We selected to express our values in absolute and normalized form for ease of comparison and to try and follow the recommended clinical practice, which may give confidence in suggesting relative and acute AKI in this athletic population. Both the absolute and normalized analyses showed an increase in uCyst-C concentration. In reference to this AKI marker, this suggests that, over the 5-day span, there was some level of AKI experienced by the athletes observed.

Notably, the time samples we collected were limited to a few practice sessions during the preseason. Thus, time sample periods may have been missed that better highlight a time course of AKI-related renal function in athletes. Additionally, only 42% of this sample athlete population was observed over this time. While the female collegiate soccer athletes we observed were considered a homogenous convenience sample, there was variation in these responses, so we are careful to generalize the outcomes seen in this study to other training athletes. It is further evident that more information is needed to fully understand the behavior of uCyst-C under these conditions for a longer time course in differing populations.

In our observational analysis of uNGAL, we found no difference (*p* = 0.40) in urinary concentration between any practice sessions and baseline values. Previous investigations that assessed uNGAL changes found a ~4–6-fold change in uNGAL concentration following a marathon. However, other previous studies showed variable changes in uNGAL in exercise bouts that were much shorter in duration than a marathon. No differences were found following an acute resistance training bout or a submaximal treadmill test [68,69], while another study showed an increase after an 800 m run [32]. Interestingly, a study investigating high-intensity interval resistance training on markers of AKI showed an increase in uNGAL after it was normalized to uCr. Two females were found near clinical AKI levels (100 ng·uCr-mg^−1^) 2 h post-bout. However, these values returned to baseline after 24 h [33]. This suggests that there may be some inter-variability in acute renal responses to exercise bouts. Overall, it appears that uNGAL increases due to longer exercise bouts (e.g., marathon), yet it rarely exceeds normal values when normalized to uCr [12]. Our data showed no changes from the baseline values during any practice sessions and was not normalized to uCr. It should be noted that the majority of urine samples were collected in the morning after a previous practice session, and similar to previous findings where uNGAL returned to baseline after 24 h, our selected sample time periods may have been enough time for uNGAL clearance. Additionally, with a small sample number (*n* = 7) and variable uNGAL responses to exercise-induced stress, this may also explain why no differences were seen. Lastly, other factors such as heat acclimatization strategies, hydration protocols used, and the time course of uNGAL metabolism are interrelated to changes found in uNGAL [12]. Therefore, interpreting uNGAL changes can be challenging. It has been suggested that the increase in uNGAL could be related to metabolic renal adaptation exercise, or possibly predisposition to AKI over time [26,70].

## 5. Conclusions

In summary, NCAA DI female soccer athletes maintained euhydration (≤1.020 g·mL^−1^) throughout preseason training, which may have mitigated kidney stress despite a high-intensity exercise in a hot, humid environment. Urinary creatinine (uCr) remained within the normal range (208 mg·dL^−1^), likely elevated due to training demands. Notably, urinary cystatin C (uCyst-C) increased to 0.42 mg·L^−1^ after the first exhibition game (D5), nearing the AKI threshold (≥0.45 mg·L^−1^), yet it subsequently declined, suggesting potential renal adaptation to training. No significant changes were observed in urinary NGAL (uNGAL), possibly due to individual variability and training bout duration. These findings suggest that uCyst-C may serve as a sensitive biomarker for workload and acute kidney stress in collegiate soccer athletes, with hydration status playing a key role in mitigating potential renal strain.

## Figures and Tables

**Figure 1 nutrients-17-02185-f001:**
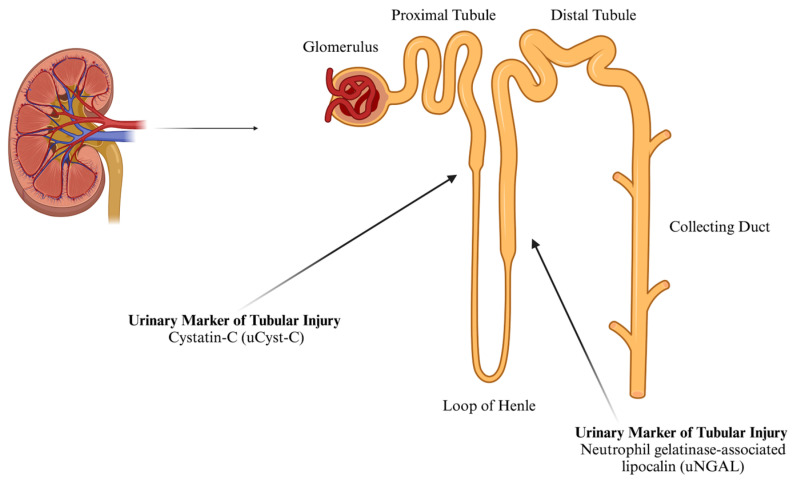
The approximate location of the synthesis of urinary markers of tubular AKI (created using BioRender).

**Figure 2 nutrients-17-02185-f002:**
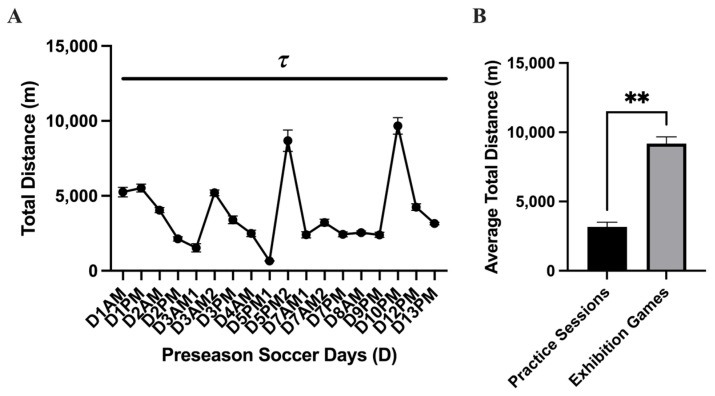
Total distance displaced (mean ± SEM): The analysis of the (**A**) total distance displaced (m) during the preseason showed a main effect time difference (*p* = <0.0001 τ) between practice sessions. Post hoc analysis showed multiple time differences between practice sessions during the preseason, which is represented by the black dash bar (

). (**B**) Comparing the average total distance during practice sessions and the exhibition games, there was a higher average total distance displaced during the exhibition games (*p* = 0.007 **).

**Figure 3 nutrients-17-02185-f003:**
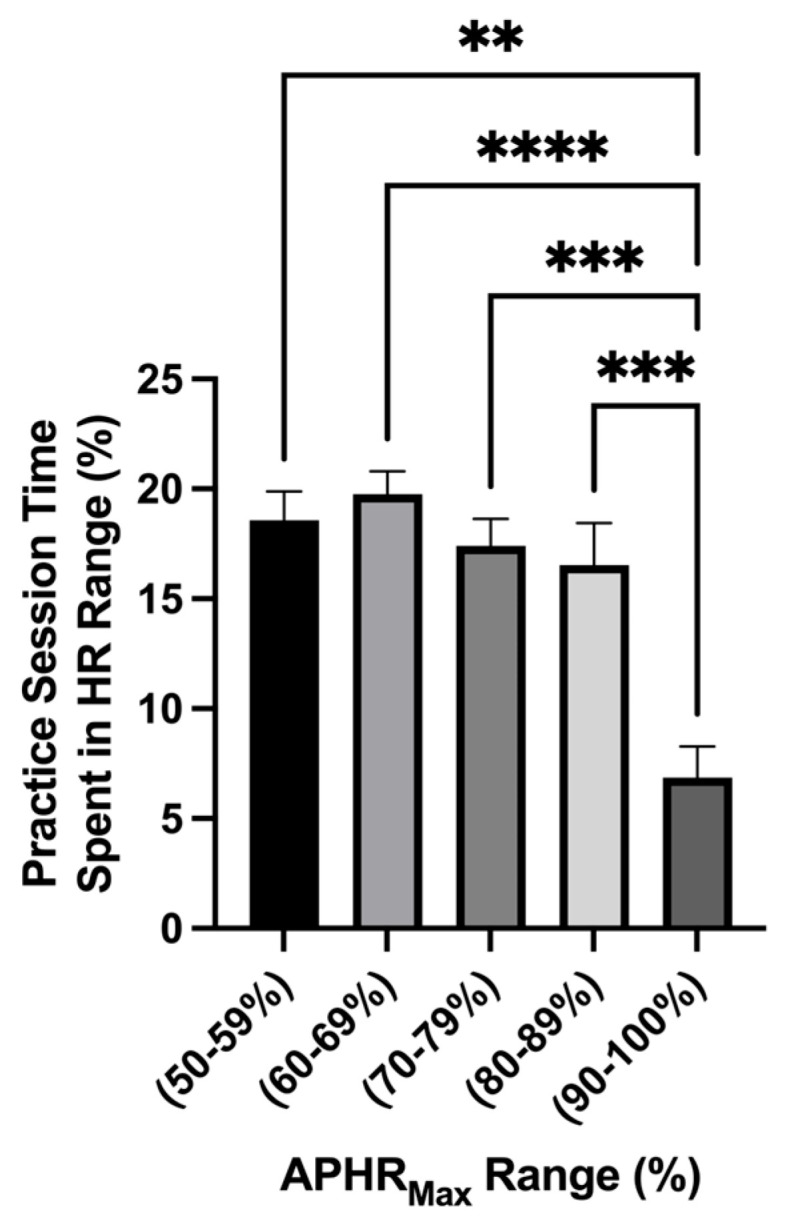
Time spent in %APHR_Max_ range (mean ± SEM): The analysis of the percentage of time spent in an age-predicted heart rate max range showed a main effect difference in time spent in the HR range (*p* = <0.0001). The time spent in the HR range 50–59% (*p* = 0.002 **), 60–69% (*p* = <0.0001 ****), 70–79% (*p* = 0.0004 ***), and 80–89% (*p* = 0.0001 ***) was greater than 90–100%.

**Figure 4 nutrients-17-02185-f004:**
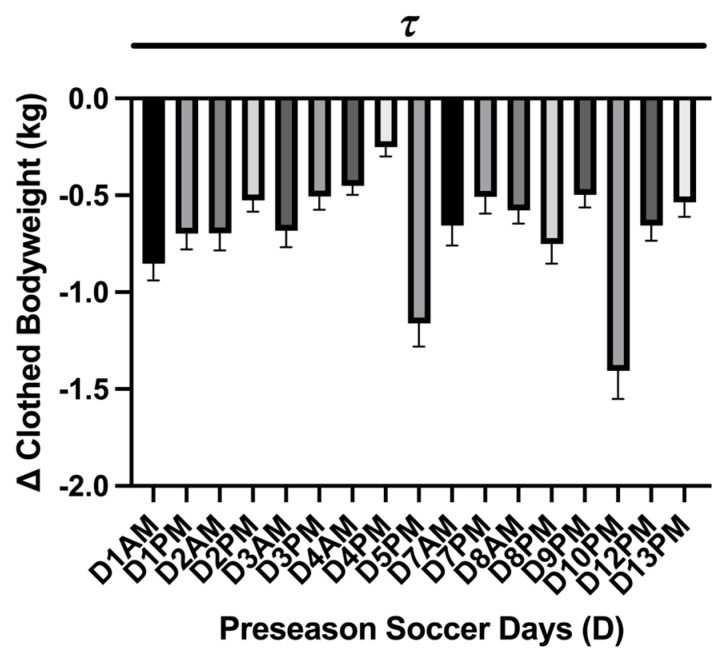
Practice session body mass change (mean ± SEM): The analysis showed a main effect of time on kg body mass loss (*p* = <0.0001 τ). Post hoc analysis showed multiple time differences between practice sessions during the preseason, which is represented by the black dash bar (

). Day (D), the practice session time of day (a.m./p.m.).

**Figure 5 nutrients-17-02185-f005:**
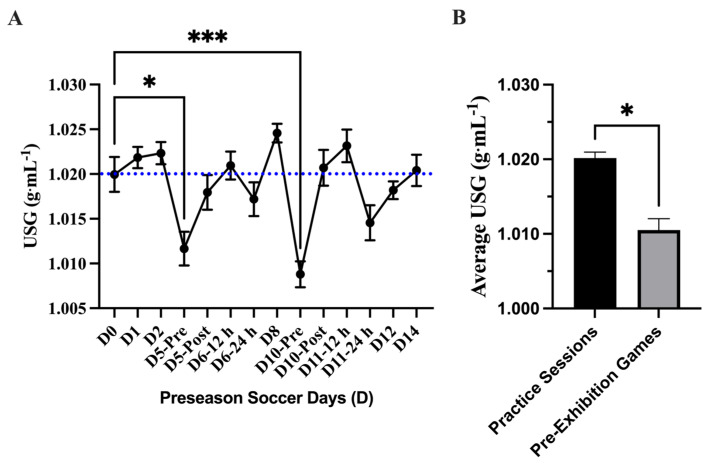
Urine specific gravity (USG) (mean ± SEM): (**A**) Analysis showed a main effect of time on USG (*p* = <0.0001). Post hoc analysis showed that D5-Pre (*p* = 0.04 *) and D10-Pre (*p* = 0.0003 ***) had lower USG values than baseline (D0). Day (D); pre, post, 12 h, and 24 h coordinate with time points related to exhibition games 1 and 2. The blue dotted line denotes the threshold of euhydration (≤1.020). (**B**) Comparing the average USG measures between the practice session and the exhibition games showed a difference (*p* = 0.04 *) and a lower average USG value for pre-exhibition game days (D5-Pre; D10-Pre).

**Figure 6 nutrients-17-02185-f006:**
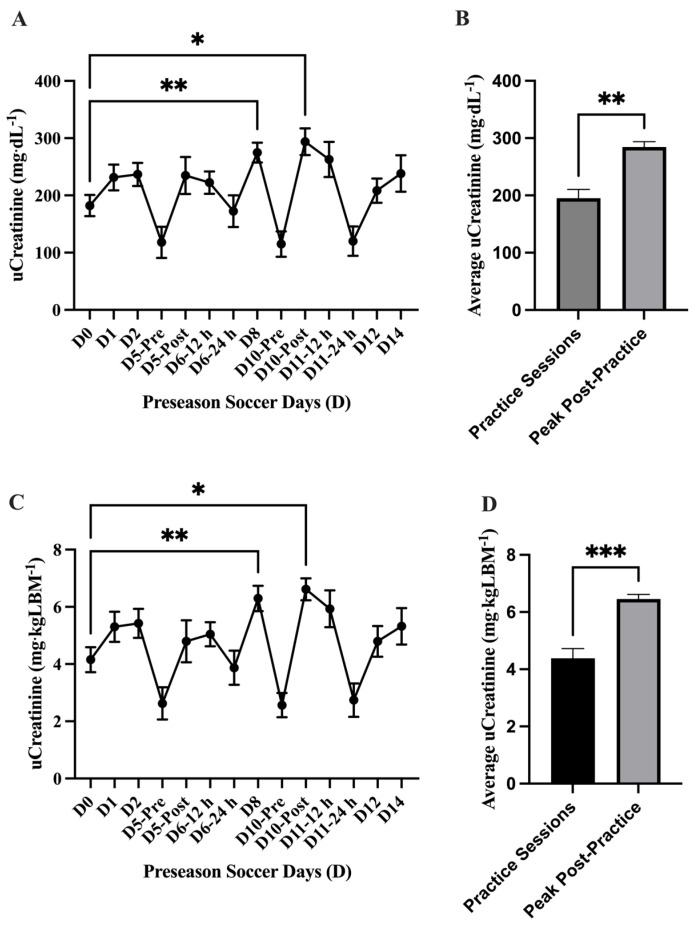
Urinary creatinine (uCr) (mean ± SEM): The analysis of (**A**) absolute uCr concentration (mg·dL^−1^) showed a main effect of time on uCr (*p* = <0.0001). Post hoc analysis showed that D8 (*p* = 0.002 **) and D10-Post (*p* = 0.02 *) had higher uCr values than baseline (D0). (**B**) The average absolute uCr concentration (mg·dL^−1^) was found higher (*p* = 0.001 **) in the practice sessions D8 and D10-Post (*n* = 2) compared to the other practice sessions (*n* = 12). The analysis of (**C**) normalized uCr (mg·dL^−1^·LBM^−1^) showed a main effect of time on uCr (*p* = <0.0001). Post hoc analysis showed that D8 (*p* = 0.001 **) and D10-Post (*p* = 0.02 *) had higher uCr values than baseline (D0). (**D**) The average normalized uCr concentration (mg·dL^−1^·LBM^−1^) was found to be higher (*p* = 0.0002 ***) in the practice sessions D8 and D10-Post (*n* = 2) compared to the other practice sessions (*n* = 12). Day (D); pre, post, 12 h, and 24 h coordinate with time points post-exhibition games 1 and 2.

**Figure 7 nutrients-17-02185-f007:**
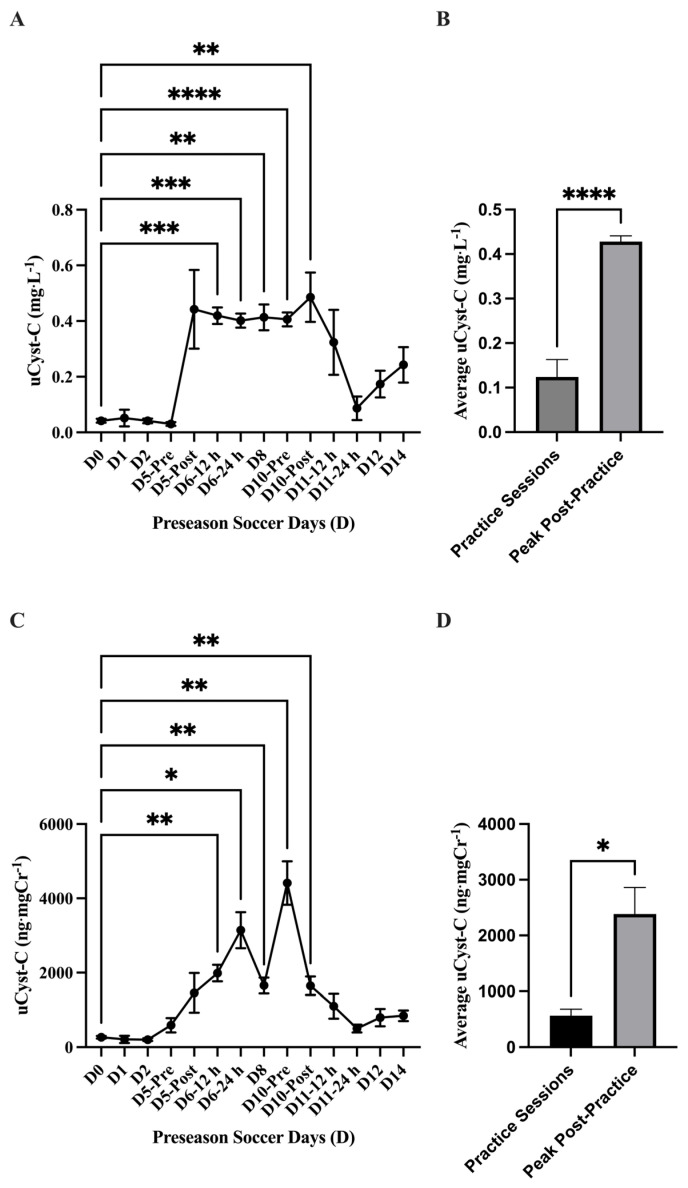
Urinary cystatin C (uCyst-C) (mean ± SEM): The analysis for (**A**) absolute uCyst-C (mg·L^−1^) concentration showed a main effect of time on uCyst-C (*p* = 0.001). Post hoc analysis showed that D6-12 h (*p* = 0.0001 ***), D6-24 h (*p* = <0.0001 ****), D8 (*p* = 0.002 **), D10-Pre (*p* = <0.0001 ****), and D10-Post (*p* = 0.008 **) had higher uCyst-C concentration values compared to baseline (D0). (**B**) The average absolute uCyst-C concentration (mg·L^−1^) was found to be higher (*p* = <0.0001 ****) in the practice sessions D5-Post-D10-Post (*n* = 6) compared to the other practice sessions (*n* = 8). (**C**) The analysis for normalized uCyst-C (ng·mgCr^−1^) showed a main effect of time (*p* = <0.0001). Post hoc analysis showed that Day 5-Post (*p* = 0.04 *), D6-12 h (*p* = 0.002 **), D6-24 h (*p* = 0.008 **), D8 (*p* = 0.003 **), D10-Pre (*p* = 0.002 **), and D10-Post (*p* = 0.004 **) had higher uCyst-C concentration values compared to baseline (D0). (**D**) The average normalized uCyst-C concentration (ng·mgCr^−1^) was found to be higher (*p* = 0.01 *) in the practice sessions D5-Post-D10-Post (*n* = 6) compared to the other practice sessions (*n* = 8). Day (D); pre, post, 12 h, and 24 h coordinate with time points related to exhibition games 1 and 2.

**Figure 8 nutrients-17-02185-f008:**
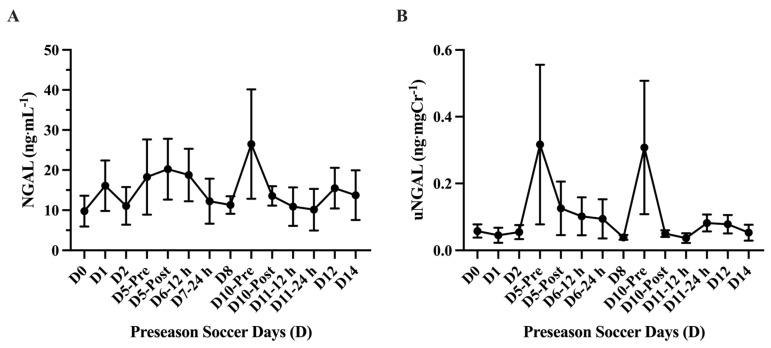
Urinary neutrophil gelatinase-associated lipocalin (uNGAL) (mean ± SEM): No differences (*p* > 0.05) were found comparing any practice day to baseline (D0) in absolute (**A**) (mg·mL^−1^) or normalized (**B**) (ng·mgCr^−1^) uNGAL concentration.

**Table 1 nutrients-17-02185-t001:** Index for hydration status.

Condition	USG Value
Well hydrated	<1.010
Minimal dehydration	<1.010–1.020
Significant dehydration	1.021–1.030
Serious dehydration	>1.030

Urine specific gravity (USG), ≤1.020, is an indication of euhydration status [38].

**Table 2 nutrients-17-02185-t002:** Preseason practice schedule.

Day	Preseason Practice Name	Time of Day (h)
D0	Pre-Preseason (Baseline)	04:30–07:00
D1	Fitness Testing Day #1	06:00–10:00 and 18:00–21:00
D2	Fitness Testing Day #2	06:00–10:00 and 18:00–21:00
D3	Regular Practice Day #1	06:00–10:00 and 18:00–21:00
D4	Regular Practice Day #2	06:00–10:00 and 18:00–21:00
D5	Exhibition Game #1	17:00–19:00
D6	Scheduled Rest Day #1	05:00–07:00 *
D7	Regular Practice Day #3	06:00–10:00 and 18:00–21:00
D8	Midweek #1—Regular Practice Day	06:00–10:00 and 18:00–21:00
D9	Regular Practice Day #4	06:00–10:00 and 18:00–21:00
D10	Exhibition Game #2	19:00–21:00
D11	Scheduled Rest Day #2	05:00–07:00 *
D12	Midweek #2—Regular Practice Day	18:00–21:00
D13	Regular Practice Day #5	18:00–21:00
D14	Post–Preseason (End)	04:30–07:00

Preseason practice schedule, name, and associated practice times. Day (D), time is expressed in a 24 h clock, each session time frame includes the practice session time and data collection; * denotes data collection only.

**Table 3 nutrients-17-02185-t003:** Descriptive statistics.

Athletes (Baseline)	Normal Reference	Mean ± SD	95% CI	*n*
Age (y)		19.3 ± 1.17	18.85–19.91	21
Height (cm)		169.6 ± 6.24	166.8–172.5	21
Body Mass (kg)		68.4 ± 11.28	54.09–99.5	21
Lean Body Mass (kg)		45.9 ± 5.18	36.41–56.01	21
Fat Mass (kg)		22.5 ± 7.75	13.91–43.58	21
Body Fat (%)		32.2 ± 6.05	24.2–44.2	21
uCyst-C (mg·L^−1^)	0.06–0.16 ♀/♂	0.041 ± 0.019	0.024–0.058	8
uNGAL (ng·mL^−1^)	≤65.0 ♀	17.65 ± 17.03	1.89–33.40	7
uCr (mg·dL^−1^)	~20–400 ♀/♂	182.3 ± 63.84	141.8–222.9	12
Urine Specific Gravity (USG)		1.020 ± 0.008	1.016–1.024	21
**Athlete Workload (Average)**		**Mean ± SD**	**95% CI**	** *n* **
Practice Session Time (min)		97.38 ± 69.68	62.73–132.0	18
Total Distance (m)		3832 ± 2344	2666–4997	18
Velocity (m·min^−1^)		44.4 ± 18.81	35.05–53.76	18
APHR_Avg_ (%)		65.02 ± 8.28	65.57–70.43	18
APHR_Max_ (%)		97.30 ± 5.08	94.77–99.82	18
**Environment (Average)**		**Mean ± SD**	**Min–Max**	** *n* **
Ambient Temperature (°C)		30.82 ± 2.03	29.87–31.77	20
Relative Humidity (%RH)		71.69 ± 8.16	67.87–75.51	20
Wet-Bulb Globe Temperature (WBGT) (°C)		27.43 ± 0.82	27.01–27.84	18

Age-predicted heart rate (APHR); urinary cystatin C (uCyst-C); urinary neutrophil gelatinase-associated lipocalin (uNGAL); urinary creatinine (uCr); female (♀), male (♂).

## Data Availability

The raw data supporting the conclusions of this article will be made available by the authors upon reasonable request due to privacy of the female athletes, who could be indirectly identified due to the university-associated information freely located online.

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
