# Peer review of "Hydration Status and Acute Kidney Injury Biomarkers in NCAA Female Soccer Athletes During Preseason Conditioning"

_nutrients, 2025, doi:10.3390/nu17132185_

Round 1

Reviewer 1 Report

Comments and Suggestions for Authors

Dear Authors,

The thematic of our article is of interest, but a great effort have to be done to organize better the presentation.  The objective of your study is to “investigate the roles of athlete workload and hydration status during multiple practice sessions on urinary biomarkers reflective of AKI during pre-season training”. Your methodology and results presentation should be organized to help future lecturer having no doubt on how you proceeded to respond to this question.

As it is, some results appear in the discussion, but it is difficult to understand from witch treatments they came from.

Your study shows significant difference between training and games in all the figures presented, indicating the possible influence of the workload on the parameter analysed. Correlations between athlete’s workload and AKI and hydration, should be of interest to provide more information’s.  For the same parameters, compare the average of the training in the first week with the average of the second week, and presenting ANOVAS between the training in each week, may help to clarify the evolution of the parameters. Probably you did part of our suggestions, but it is not presented in your results and in the discussion, it remains unclear. Please improve those main aspects.

Correlations with ambient parameters are also a possibility to analyse.

Additional recommendations:

You use weight in your terminology, but body mass is probably more appropriate, especially in results tables.  

In material and methods, in 2.3., for clothed body mass it is necessary provide more details and appropriate reference for the procedure, 36 do not explain this procedure. It is important to explain if before and after the training the subject were weighted with the same clothes or if after the training they had to change for the same dry clothes to avoid including accumulated sweat in the final weigh-in. It is also important to refer if the drink during the training were controlled to estimate the effective weight loss by sweating during the effort. If those precautions were not used it is necessary to justify why your procedure remain valid.

In 2.5., it is necessary to specify witch kind of velocity (we suppose it is average velocity), for age predicted heart rate, you should specify witch formula was used and put some reference.

Fitness test day, maybe you should explain shortly witch kink of test and refer how it was possible to obtain distance and speed results from those tests.

It is important to refer if during the trainings and games, cooling strategies like use of ice, fan, … were used to prevent heat stroke. And discuss the eventual influence of those factors on the results of your study.

Lines 247 and 248, this sentence is not clear, please explain better.

2.8. and 2.9. please include references for the procedures used.

2.10. statistical procedure used deserve to be reformulated with references and clear explanations about the usefulness of those procedures to deserve the objectives of the study.   

Table 3. You did not provide explanation about why for AKI parameter the number of subjects was 8, 7, and 12, while the number of subjects was 21. Please clarify somewhere in the article.

About anthropometric and body composition data’s, you have those results in D0 and D14, why not analysing the evolution of those parameters?

Line 331, it is probably figure 2 and not table 2.

Author Response

As it is, some results appear in the discussion, but it is difficult to understand from witch treatments they came from.

Thank you for this feedback. Are there examples that the reviewer could give so we are able to remedy it? We are having a difficult time discerning the reviewer’s request.

Your study shows significant difference between training and games in all the figures presented, indicating the possible influence of the workload on the parameter analysed.

Thank you for this feedback. We did not find a correlation between workload and environment with our outcome measures. While there were numerous differences found for distance (workload), this did not correlate with measures of AKI and USG.

Correlations between athlete’s workload and AKI and hydration, should be of interest to provide more information’s.  

We ran a correlational analysis for the factors listed above (i.e., Distance, uCyst-C, and USG) and found very weak r values and/or no significance. We will add additional language to the discussion, suggesting no correlation was found between workload and the markers of AKI and hydration. Additionally, we have added language in 2.10, the statistical methods that highlights the use of correlational analysis. 

For the same parameters, compare the average of the training in the first week with the average of the second week, and presenting ANOVAS between the training in each week, may help to clarify the evolution of the parameters. Probably you did part of our suggestions, but it is not presented in your results and in the discussion, it remains unclear. Please improve those main aspects.

Thank you for this feedback, the reviewer is correct. We did find differences in distance in each practice session. However, no correlations were found between training workloads and our AKI and hydration outcome measures. We will add language to address this in the discussion.

Correlations with ambient parameters are also a possibility to analyse.

Thank you for this feedback. We did not find any correlations between environmental factors and our outcome measures, unfortunately. This may be due to less control in the field-based study or the time of exposure to the environment, and the acclimation status of the athletes was unknown.  

Additional recommendations:

You use weight in your terminology, but body mass is probably more appropriate, especially in results tables.

Thank you for this feedback, we agree with the reviewer and have changed this wording.   

In material and methods, in 2.3., for clothed body mass it is necessary provide more details and appropriate reference for the procedure, 36 do not explain this procedure. It is important to explain if before and after the training the subject were weighted with the same clothes or if after the training they had to change for the same dry clothes to avoid including accumulated sweat in the final weigh-in. It is also important to refer if the drink during the training were controlled to estimate the effective weight loss by sweating during the effort. If those precautions were not used it is necessary to justify why your procedure remain valid.

Thank you for the feedback. We have added additional language to this section to highlight the limitations of our study design. Additionally, we reiterated this limitation in our discussion section to highlight this limitation. We would also like to inform the reviewer that this was a female soccer team, and this limited any male research participation in the locker room to assist with urine collection and pre-/post-session body mass measures.

“Athletes were instructed to wear the same clothing for their pre- and post-practice session body mass measures. Perspiration that may have accumulated in clothing post-practice session was not controlled. For this study, hypohydration was defined as USG >1.020 [37] and > 2 % of body mass loss [36]. The index for the range of hydration status for USG is listed below for reference in Table 1. Additionally, water ingestion during practice sessions was not controlled or monitored due to the difficulty of observing the daily intake of water in multiple athletes (n=21) over the pre-season time period.”

“However, it should be highlighted that previous work controlled for fluid intake, urine output, and estimated sweat rate [53]. Additionally, due to the inherent ‘field-based’ study design, we opted to collect non-nude body mass, and the athletes were not instructed to change into dry clothes or towel off excess perspiration post-practice sessions, which has been a recommended protocol to more accurately measure body mass changes [54].”

In 2.5., it is necessary to specify witch kind of velocity (we suppose it is average velocity), for age age-predicted heart rate, you should specify witch formula was used and put some reference.

Thank you for this feedback. The Polar Pro system is inherently programmed with the inaccurate 220-age formula (https://support.polar.com/e_manuals/Team_Pro/Polar_Team_Pro_user_manual_English/manual.pdf), and we have now highlighted this in that section. Additionally, we did express avg velocity and placed that term appropriately.

“However, it should be highlighted that this system inherently uses the commonly used “HRMax=220-Age”, which has been suggested to be inaccurate with a standard deviation of 10-12 bpm and has been shown to underestimate HRMax in younger and older adults [44]. Workload metrics to highlight the subject workload per session were total distance (m), average velocity (m⋅min-1), age-predicted heart rate average (APHRAvg %), and max (APHRMax %). The Polar Team Pro System was owned and operated by the Athletics Department.”

We also added this additional language to the discussion section to highlight the prediction equation limitation: “However, we did not find an association between distance workload and outcome measures of AKI and USG. Additionally, due to the inherent use of APHRMAX of 220-age by Polar Team Pro that was used to predict HRMax for the athletes and its known inaccuracies [44], great caution should be used when interpreting and generalizing the HR results of this study. Concurrent with the limitations of the inherent HRMAX prediction equation,”  

Fitness test day, maybe you should explain shortly witch kink of test and refer how it was possible to obtain distance and speed results from those tests.

Thank you for the feedback. “Fitness testing day” consisted of 2 sessions per day. Fitness Testing Day #1 consisted of a 2-mile run in the morning, followed by a practice session later that day. Fitness Testing Day #2 consisted of the Beep Test in the morning and a regular practice session in the evening. They wore GPS trackers both of these days, also.

It is important to refer if during the trainings and games, cooling strategies like use of ice, fan, … were used to prevent heat stroke. And discuss the eventual influence of those factors on the results of your study.

We appreciate the feedback. However, those interventions were not part of the main outcome measures of the study. We were not looking at the incidence of heat stroke/other EHIs therefore therefore that is why those items were omitted. Ice and fanning are helpful pre-cooling strategies to assist with thermal comfort, however, the aims of this study relate to acute kidney injury- therefore, cooling strategies were not applicable to this intervention.

Lines 247 and 248, this sentence is not clear, please explain better.

Thank you, we have altered the paragraph and hope this reads more clearly: “Athletes were provided with a urine specimen cup and directed to the nearest restroom to the laboratory. After baseline collections on Day 0 (D0), preseason training began the following morning around 0700. This included the start of fitness testing assessments. Fitness testing assessments were conducted over the first two days of preseason (Days 1 and 2; D1 & D2). Urine samples were collected each morning between 0500 and 0630 throughout the preseason training period.

2.8. and 2.9. please include references for the procedures used.

We have added references to these sections.

2.10. statistical procedure used deserve to be reformulated with references and clear explanations about the usefulness of those procedures to deserve the objectives of the study.

We appreciate your feedback; however, we are having a difficult time discerning the reviewer’s request above. I believe we clearly stated that the primary objective of this field-based study was to observe hydration status and urinary AKI markers over the pre-season period in female soccer athletes. Unfortunately, we did not find any association with distance/environment and the outcome measures to report. This is likely due to a lack of control over daily and practice session water intake, which is confounded by body mass changes resulting from perspiration. However, we would like to highlight that this was a field-based sport study that the main objective was to observe renal function measures in this population. If we are interpreting the reviewer’s request incorrectly, please assist us better with a clearer request.

Table 3. You did not provide explanation about why for AKI parameter the number of subjects was 8, 7, and 12, while the number of subjects was 21. Please clarify somewhere in the article.

Thank you for this feedback. We did explain this in section 2.9: “The athletes with the highest workload ratio (m⋅min-1⋅HRAvg-1) collected from the Polar Monitor Pro System were selected for urinary analysis for AKI markers. It was assumed that these athletes would have had the highest stress placed on their inherent renal system due to higher average workloads during the preseason.” Additionally, the ELISA costs associated with multiple time points and athletes were not permissible due to limited funding for urinary analysis.

About anthropometric and body composition data’s, you have those results in D0 and D14, why not analysing the evolution of those parameters?

We appreciate the reviewer’s desire to identify/highlight all other changes in this study design, but compositional changes were not our primary objective. Our main objective was to investigate urinary AKI measures indicative of renal function in female collegiate athletes.

Line 331, it is probably figure 2 and not table 2.

Thank you for finding this. You are correct. We have altered the text as suggested.

Reviewer 2 Report

Comments and Suggestions for Authors

The manuscript is on an important topic and focuses on female athletes who are traditionally under-studied compared to males. The manuscript is generally well-written.

The study is highly original, as I could only find one other study assessing kidney function in soccer players in response to training or game play (https://pubmed.ncbi.nlm.nih.gov/25270786/). I suggest mentioning this in your manuscript discussion.

If there is space, I suggest mentioning the mean environmental heat conditions (from lines 210-211) in the abstract.

Is there a hypothesis that you could propose at the end of the introduction?

Line 167: “Hydration status was measured utilizing changes in clothed-body mass pre- and post-practice sessions”. Most of the body mass loss during a dehydrating training session will be through sweat loss. Much of the sweat will be absorbed on the athlete clothing. Please comment on this limitation.

Results: Table 3: Please clarify that these are baseline values. Some of the participants at baseline had “significant dehydration” according to the 95% CI for urine specific gravity. How many participants had significant dehydration?

Line 331: I think this table might be missing- or should this refer to table 3 instead?

When describing the p-values in brackets throughout the results section, the “****” are unnecessary.

Line 364: In this paragraph you describe body weight changes (losses) in absolute terms (kg). Since dehydration is usually considered at 2% body weight, I think you should present your data in relative terms instead (% body weight loss).

Line 477: “Previous reports observing soccer athlete displacement during women’s soccer game matches were ~ 5.4 [47] and 5.6 km [48].” Other assessments of displacement during women’s soccer games (using GPS technology) indicate a displacement of about 9 km (https://pubmed.ncbi.nlm.nih.gov/32384719/) which is closer to values in your study.

Lines 548-549: I suggest changing the term “injury” to “damage” here. Muscle injury and damage are different phenomena.

Line 559: “...a reduction in creatine phosphagen turnover and metabolism from high-intensity practice sessions”. Should this be “increased” instead of “reduction”. Use of creatine phosphate would increase during high-intensity sessions.

Did you assess whether any participants were taking creatine supplements as this might affect urinary creatinine values?

Line 619: Is “mixed with” the correct wording intended here?

Author Response

The study is highly original, as I could only find one other study assessing kidney function in soccer players in response to training or game play (https://pubmed.ncbi.nlm.nih.gov/25270786/). I suggest mentioning this in your manuscript discussion.

Thank you for this feedback, we will add additional language to our discussion, citing this article and the limited work done on kidney function in soccer players. It should read:

“To our knowledge, only one previous study has assessed renal function in soccer players post-match play [46]. This is the first study to follow and examine multiple practice sessions and the hydration status of female soccer athletes and how this may impact urinary-derived biomarkers associated with AKI.”

If there is space, I suggest mentioning the mean environmental heat conditions (from lines 210-211) in the abstract.

Thank you, we have added to the abstract and it should read “The average temperature was 27.43±0.19 °C, and the humidity was 71.69±1.82 %.”

Is there a hypothesis that you could propose at the end of the introduction?

We added a generalized hypothesis statement that should read: “We hypothesized that the concurrent effect of higher temperatures and humidity, and the assumed pre-season training status of the athletes, may negatively affect the markers of hydration and AKI.”

Line 167: “Hydration status was measured utilizing changes in clothed-body mass pre- and post-practice sessions”. Most of the body mass loss during a dehydrating training session will be through sweat loss. Much of the sweat will be absorbed on the athlete clothing. Please comment on this limitation.

The reviewer is correct and highlights one of many limitations of this study design. We have added language to this section and the discussion to highlight this limitation and direct the reader to be cautious of generalizing our outcomes and limits our confidence of using body mass changes as a reference for hydration.

“Athletes were instructed to wear the same clothing for their pre- and post-practice session body mass measures. Perspiration that may have accumulated in clothing post-practice session was not controlled. For this study, hypohydration was defined as USG >1.020 [37] and > 2 % of body mass loss [36]. The index for the range of hydration status for USG is listed below for reference in Table 1. Additionally, water ingestion during practice sessions was not controlled or monitored due to the difficulty of observing the daily intake of water in multiple athletes (n=21) over the pre-season time period.”

However, it should be highlighted that previous work controlled for fluid intake, urine output, and estimated sweat rate [53]. Additionally, due to the inherent ‘field-based’ study design, we opted to collect non-nude body mass, and the athletes were not instructed to change into dry clothes or towel off excess perspiration post-practice sessions, which has been a recommended protocol to more accurately measure body mass changes [54].”

Results: Table 3: Please clarify that these are baseline values. Some of the participants at baseline had “significant dehydration” according to the 95% CI for urine specific gravity. How many participants had significant dehydration?

The reviewer brings up a good point. According to our data, n=10 individuals had a USG measure of >1.020. However, many of our USG measures were in the morning prior to practice, and we did not control for water intake. It is unknown if the hydration status was related to a lack of water ingestion prior to sample collection or a normal response from 8-12 h of sleeping with limited water intake overnight.  

Line 331: I think this table might be missing- or should this refer to table 3 instead?

Thank you for finding this. It should state “Figure 2” as opposed to Table 2. We have altered this term.

When describing the p-values in brackets throughout the results section, the “****” are unnecessary.

Thank you for the feedback. While we agree with the reviewer and it can be perceived as redundant, the **** denotes the p-value that GraphPad produces for its figures. We maintained this process to be consistent in case a reviewer inquired what the asterisk denotes, ‘magnitude of p value difference’ from previous review experiences.

Line 364: In this paragraph you describe body weight changes (losses) in absolute terms (kg). Since dehydration is usually considered at 2% body weight, I think you should present your data in relative terms instead (% body weight loss).

We 100% agree with the reviewer’s comments and feel that the relative change in body mass is a more accurate expression of the data. However, the body mass data for this study is no longer accessible to analyze (collected at a different university in 2019). We were only able to express the kg change from pre to post practice session. Additionally, because we did not control for the impact of sweat saturation in clothing, measure nude bodyweight, and we did not control for water intake due to this being a field study, even if we were able to express this data, we would be limited to making any definitive conclusions. We have added additional language to the discussion section to highlight these limitations.

Line 477: “Previous reports observing soccer athlete displacement during women’s soccer game matches were ~ 5.4 [47] and 5.6 km [48].” Other assessments of displacement during women’s soccer games (using GPS technology) indicate a displacement of about 9 km (https://pubmed.ncbi.nlm.nih.gov/32384719/) which is closer to values in your study.

Thank you for this feedback and reference. We have added additional language with this reference to the same area in the discussion.

Lines 548-549: I suggest changing the term “injury” to “damage” here. Muscle injury and damage are different phenomena.

 Thank you for this feedback, we have altered the term used.

Line 559: “...a reduction in creatine phosphagen turnover and metabolism from high-intensity practice sessions”. Should this be “increased” instead of “reduction”. Use of creatine phosphate would increase during high-intensity sessions.

Thank you, great eyes! We have altered from reduction to an increase.

Did you assess whether any participants were taking creatine supplements, as this might affect urinary creatinine values?

We did not collect any dietary/supplementary data for this study. This is indeed a limitation. We have added language to Discussion in 4.2 that should read "Lastly, it should be noted that daily dietary and water intake were not recorded for this ‘field-based’ study. Due to this limitation, we are unable to isolate and explain how that may influence hydration status in these athletes."

Line 619: Is “mixed with” the correct wording intended here?

Thanks for finding this. We have removed the incorrect wording.

Reviewer 3 Report

Comments and Suggestions for Authors

This is an interesting study that provides new information on hydration in team sports, particularly women's football. The authors have clearly presented their work, and the study is interesting and enjoyable for readers.

However, to enhance the credibility of Nutrients, I believe that important information is missing from the methodology and statistical analysis. As a general comment, I note that fluid intake during training or games, along with daily fluid intake, has not been mentioned. I also believe that the authors should mention the daily variations in temperature and humidity, not just the averages. I provide specialized comments below.

  • The term "acute" refers to periods of hours and days, while the study concerns the monitoring of a mesocycle.
  • There are common words in the keywords and the title.
  • I didn’t see the Ethics committee reference number
  • Perhaps the most important issue with the methodology is that neither fluid (water) intake during training (which could be controlled by the researchers) nor daily fluid intake is mentioned anywhere.
  • Regarding the results, calculating training loads at the microcycle level and correlating them with AKI parameters may provide more insight into hydration and training load. Indeed, this is a new element that will be added to the literature.
  • Additionally, providing information on daily temperature and humidity rather than averages would be informative.
  • A regression analysis that includes environmental conditions, training load, and hydration would be particularly significant for interpretation through physiology.

Author Response

  • The term "acute" refers to periods of hours and days, while the study concerns the monitoring of a mesocycle.

We are unsure what the reviewer is referring to in the manuscript. If the reviewer is highlighting the use of acute for “Acute Kidney Injury”, this is a condition that has been well established (https://www.ncbi.nlm.nih.gov/books/NBK441896/). The total time of observation was ~14 days.

  • There are common words in the keywords and the title.

We have deleted the redundant use of ‘hydration status” that is used in the title, but maintained AKI due to specific outcome measures related to that condition

  • I didn’t see the Ethics committee reference number

The IRB reference number has been added for the reviewer’s convenience.

  • Perhaps the most important issue with the methodology is that neither fluid (water) intake during training (which could be controlled by the researchers) nor daily fluid intake is mentioned anywhere.

This is indeed a limitation to this “field” study design. Due to observing n=21 athletes, we elected not to control for daily water ingestion and dietary intake. We placed a statement in the discussion highlighting this limitation, and therefore, limits what we can conclude with this study. “However, it should be highlighted that this group controlled for fluid intake, urine output, and estimated sweat rate. Because we did not control for these factors, we collected non-nude body mass; we are unable to definitively suggest that these athletes were within a normal fluctuation of body water.”

  • Regarding the results, calculating training loads at the microcycle level and correlating them with AKI parameters may provide more insight into hydration and training load. Indeed, this is a new element that will be added to the literature.

Thank you for this feedback. We have run correlations with workload (distance) and AKI/USG measures and found either no significance or a very weak correlation (r = .26). This is probably best explained by the lack of internal validity due to not controlling for water intake/total urine output, and dry, nude body weight measures. We have added additional language in the discussion highlighting the use of correlational analysis in the methods/statistical analysis section and in the discussion suggesting we found no correlation between workload and outcomes measures of hydration and AKI.    

  • Additionally, providing information on daily temperature and humidity rather than averages would be informative.

Thank you for the feedback. We did not express the daily temperature, humidity, and heat index because we did not find much variation in the time period we recorded the data. We found a CV% of ≤10% for these factors. 1SD was = ~2 ℃ difference in day-to-day temperatures. The environment was fairly stable.

  • A regression analysis that includes environmental conditions, training load, and hydration would be particularly significant for interpretation through physiology.

Thank you for this feedback. We additionally ran correlations using environmental conditions and did not find any meaningful relationships (weak r values and non-significant). This is probably explained by the lack of control for some factors, such as daily water/practice session water ingestion, clothing perspiration, which may have limited the ability to see greater changes in body mass.

Round 2

Reviewer 1 Report

Comments and Suggestions for Authors

Dear Authors,

thank you for your extensive revision of the document, it is now clear to understand, even for the statistical procedures used.

We wish success 

Reviewer 3 Report

Comments and Suggestions for Authors

The authors made a significant effort to address all of my comments. 

I suggest including the reference number and the date of the ethical committee approval. 

Kind regards